# Non-Invasive Nanometer Resolution Assessment of Cell–Soft Hydrogel System Mechanical Properties by Scanning Ion Conductance Microscopy

**DOI:** 10.3390/ijms252413479

**Published:** 2024-12-16

**Authors:** Tatiana N. Tikhonova, Anastasia V. Barkovaya, Yuri M. Efremov, Vugara V. Mamed-Nabizade, Vasilii S. Kolmogorov, Peter S. Timashev, Nikolay N. Sysoev, Victor V. Fadeev, Petr V. Gorelkin, Lihi Adler-Abramovich, Alexander S. Erofeev, Evgeny A. Shirshin

**Affiliations:** 1Department of Physics, M.V. Lomonosov Moscow State University, 1/2 Leninskie Gory, 119991 Moscow, Russia; anastasia.bark18@gmail.com (A.V.B.); nn.sysoev@physics.msu.ru (N.N.S.); victor_fadeev@mail.ru (V.V.F.); 2Institute for Regenerative Medicine, Sechenov University, 8-2 Trubetskaya St., 119991 Moscow, Russia; efremov_yu_m@staff.sechenov.ru (Y.M.E.); timashev_p_s@staff.sechenov.ru (P.S.T.); 3Laboratory of Biophysics, National University of Science and Technology MISIS, 4 Leninskiy prospekt, 119049 Moscow, Russia; vugara2003@yandex.ru (V.V.M.-N.); vskolmogorov@gmail.com (V.S.K.); peter.gorelkin@gmail.com (P.V.G.); erofeev.as@misis.ru (A.S.E.); 4Department of Chemistry, M.V. Lomonosov Moscow State University, 1/2 Leninskie Gory, 119991 Moscow, Russia; 5World-Class Research Center “Digital Biodesign and Personalized Healthcare”, Sechenov First Moscow State Medical University, 8-2 Trubetskaya St., 119991 Moscow, Russia; 6Department of Oral Biology, The Goldschleger School of Dental Medicine, Faculty of Medical & Health Sciences, Tel Aviv University, Tel Aviv 6997801, Israel; lihiab@gmail.com; 7The Center for Nanoscience and Nanotechnology, Tel Aviv University, Tel Aviv 6997801, Israel; 8The Center for the Physics and Chemistry of Living Systems, Tel Aviv University, Tel Aviv 6997801, Israel

**Keywords:** hydrogel, scaffolds, regenerative medicine, scanning ion conductance microscopy, mechanical properties, MCF-7 breast cancer cells, cell

## Abstract

Biomimetic hydrogels have garnered increased interest due to their considerable potential for use in various fields, such as tissue engineering, 3D cell cultivation, and drug delivery. The primary challenge for applying hydrogels in tissue engineering is accurately evaluating their mechanical characteristics. In this context, we propose a method using scanning ion conductance microscopy (SICM) to determine the rigidity of living human breast cancer cells MCF-7 cells grown on a soft, self-assembled Fmoc-FF peptide hydrogel. Moreover, it is demonstrated that the map of Young’s modulus distribution obtained by the SICM method allows for determining the core location. The Young’s modules for MCF-7 cells decrease with the substrate stiffening, with values of 1050 Pa, 835 Pa, and 600 Pa measured on a Petri dish, Fmoc-FF hydrogel, and Fmoc-FF/chitosan hydrogel, respectively. A comparative analysis of the SICM results and the data obtained by atomic force microscopy was in good agreement, allowing for the use of a composite cell–substrate model (CoCS) to evaluate the ‘soft substrate effect’. Using the CoCS model allowed us to conclude that the MCF-7 softening was due to the cells’ mechanical properties variations due to cytoskeletal changes. This research provides immediate insights into changes in cell mechanical properties resulting from different soft scaffold substrates.

## 1. Introduction

Over the past few years, hydrogels have emerged as promising candidates for diverse biomedical applications, including drug delivery, wound healing, and tissue engineering [1,2,3,4,5]. Specifically, hydrogels have been developed to serve as a cell-culture platform, since they effectively mimic the extracellular matrix and provide a conducive environment for cell growth and differentiation [6,7,8]. Due to their high water content and tunable mechanical properties, hydrogels have found extensive applications in regenerative medicine [9]. The stiffness and flexibility of hydrogels can be adjusted to mimic various tissue types, from the soft brain tissue, with a characteristic Young’s modulus of *E* ~ 0.1–1 kPa [10] and adipose tissues, *E* ~ 1–2 kPa [11], to the stiffer cardiac *E* ~ 30–60 kPa [12] and bone tissues E~ tens of MPa to GPa [13,14].

Hydrogels can be broadly categorized into synthetic and natural materials [15]. Synthetic hydrogels, engineered from polymers such as polyethene glycol and polyamides offer the advantage of tunability, where their mechanical properties can be precisely controlled by altering the crosslinking density [15]. Additionally, various factors such as amino acids, peptides, and glycans can be incorporated into synthetic hydrogels to improve their function. However, it has been shown that cell differentiation occurs differently on stiff synthetic hydrogels compared with a natural matrix. While providing a three-dimensional environment for cell culture, rigid synthetic scaffolds cannot activate integrins and other surface receptors. In contrast, natural hydrogels contain many binding sites for integrins and growth factors that promote proper cell differentiation [16]. Moreover, natural hydrogels derived from biological sources such as collagen [17,18], elastin [19], and fibrin [20] offer the advantage of biocompatibility and bioactivity, often promoting better cell adhesion and proliferation.

Recently, the use of Fmoc-FF peptide-based hydrogel has gained attention in the field of regenerative medicine [9,21,22,23]. This hydrogel, comprising self-assembled peptide nanofibers, presents an interesting platform for cell culturing due to its unique properties [24,25]. The softness of the Fmoc-FF hydrogel closely mimics the mechanical properties of natural tissues, making it a promising candidate for various tissue engineering applications.

There are numerous studies utilizing substrates with various topographies [26], chemical compositions [27], scales [28], and stiffnesses [29] aimed at different tissues, cells, and cellular responses. Deciphering the mechanisms and measuring the extent of influence of each of these parameters on cell populations is a complex task, as different substrate factors can influence each other. For instance, the chemical composition of substrates directly affects their stiffness [30], which in turn influences cell spreading. Consequently, cell spreading leads to high cell survival and viability.

The mechanical properties of the cell–substrate system have been the focus of numerous studies, given their direct influence on cell behavior. Such studies may include stages like obtaining cell topography on the substrate using probe or electron microscopy, obtaining the average cell stiffness in the cell–substrate system using atomic force microscopy (AFM), rheology, etc., and investigating changes in individual cell organelles during adhesion on the substrate using confocal microscopy [31,32,33,34,35], etc.

In recent years, the exploration of the mechanical properties of materials at the nanoscale has garnered significant attention, leading to the development and application of various techniques in addition to the traditional, well-recognized AFM method. Among these, optical tweezers, as reviewed by Magazzù et al. [36], utilize highly focused laser beams to manipulate and measure the mechanical properties of soft matter at the nanoscale. For instance, optical tweezers have been used to study changes in the characteristic elasticity of cells associated with various human diseases [37,38]. This technique is particularly advantageous for its non-invasive nature and high precision; however, it is effective only for manipulating very small objects, such as microparticles and molecules, which limits its application to larger samples. Additionally, the method’s effectiveness decreases when working with thicker or opaque samples. Methods are also being developed that can be used directly in clinical settings to determine tissue stiffness in patients with diseases such as prostate cancer [39]. One such method is magnetic resonance elastography. Although this technique offers a non-invasive imaging method that measures the mechanical properties of tissues, making it valuable for medical applications, it has lower spatial resolution compared with methods like optical tweezers or AFM, which may limit its application for studying small structures.

Here, we aim to provide a comprehensive overview of the mechanical properties of the cell–hydrogel system, with a particular focus on another prominent method—the use of scanning ion conductance microscopy (SICM). This technique is a powerful tool for investigating mechanical properties, offering high-resolution cell imaging and quantitative analysis [40,41,42]. In this study, MCF-7 breast cancer cells were chosen as a convenient and well-researched model for investigating the mechanical properties of the cell–soft hydrogel system using a novel alternative method, SICM. For instance, in study [43], the morphology of MCF-7 cell membranes was examined in relation to substrate stiffness, specifically changing the number of microvesicles, microvilli, and the structure of filopodia. It was hypothesized that if this convenient model could show that the SICM method allowed for a deep characterization of the cell–soft hydrogel system, with the substrate stiffness being lower than that of the cells, and that the self-assembling Fmoc-FF hydrogel would be a promising biocompatible scaffold for these cells, then this method could be used in the future to study various epithelial/neural or other cell types. This approach can examine cell–hydrogel system dynamics, allowing for the evaluation of cell mechanical characteristics following treatment with drugs and growth factors or hydrogel degradation. The novelty of this study lies, firstly, in obtaining a mechanical property distribution map of live cells on Fmoc-FF hydrogel, which can be considered a potential material for use as an extracellular matrix (ECM). Furthermore, to the best of our knowledge, there are no studies dedicated to obtaining a mechanical property map in a cell–substrate system where the substrate is softer than the cell.

## 2. Results

### 2.1. The Mechanical Properties of the Cell–Hydrogel System Assessed Using SICM

The topography and Young’s modulus distribution of low aggressive human breast cancer cells MCF-7 placed on the Petri dish and self-assembled soft Fmoc-FF hydrogel are presented in Figure 1. The investigations were carried out by SICM, and the mechanical properties were calculated via disjoining pressure. The topography of MCF-7 cells exhibited a typical epithelial-like structure with cell–cell contacts in the Petri dish, with various microvilli on their surface (Figure 1A). This observation agrees with SEM investigations [43]. The quantitative mechanical cell mapping of cells’ mechanical properties is presented in (Figure 1B); the average Young’s modulus value was *E*_mean_ = 1050 ± 55 Pa. The topography of cells placed on the soft hydrogel is presented in Figure 1C.

In this work, for the first time, the mechanical properties distribution for a cell–soft hydrogel system in an aqueous solution with nanometer resolution are presented (Figure 1D). In the inset of Figure 1C, the morphology of soft Fmoc-FF hydrogel can be seen (Figure 1E): the biomaterial comprises fibers with a characteristic length from several to tens of micrometers and with a fiber’s width ~ 150–250 nm. These data are in good agreement with confocal fluorescence microscopy (Appendix A).

Comparing cell topography on a Petri dish and the hydrogel (Figure 1A,C), correspondingly, it can be seen that MCF-7 cells shape varies when being spread on the stiff Petri dish, and cells are characterized by a rounded shape when placed on the soft hydrogel. The profiles of the cells placed on the Petri dish and Fmoc-FF hydrogel can be seen in Appendix A. Moreover, the microvilli that were clearly seen on the rigid substrate (Petri dish) disappeared (or greatly diminished in number) on the soft hydrogel. It should be noted that the images (Figure 1A,C) were obtained with the same resolution, using the capillaries with the *R_cap_* = 45 nm, so the observed changes were due to the cells’ structure variations. Microvilli are finger-like protrusions whose shape is controlled by the actin bundles [44]. Franchi et al. demonstrated by SEM analysis that, for MCF-7 cells, the densely packed collagen fibers that served as a substrate for cells stimulated the development of many cytoplasmic microvilli in contrast with low-concentrated type I collagen-covered substrates, which is in agreement with our observations.

Moreover, cells were also placed on the Fmoc-FF hydrogel, where the low molecular chitosan was added. Chitosan, being a polysaccharide extraction from natural chitin, has attracted attention in the field of tissue engineering due to its ability to increase cell viability [45,46,47], biocompatibility, and biodegradability [48]. The SICM topography and the distribution of Young’s modulus images of MCF-7 cells placed on Fmoc-FF hydrogel+chitosan can be seen in Figure 1F,G, correspondingly.

More than 40 cells were processed for each sample. Figure 1H shows the typical histogram of Young’s modulus distributions for MCF-7 cells placed on a cultural Petri dish (control, black lines), Fmoc-FF hydrogel (red lines), and Fmoc-FF+chitosan hydrogel. It should be noted that the distribution that corresponds to the MCF-7 on the Fmoc-FF and Fmoc-FF+chitosan hydrogel system (red and violet lines) had a bimodal character originated from two components: the MCF-7 cells and hydrogel. To clarify the bimodal nature of the system, Figure 1H was divided into three separate figures and also presented in Appendix A.

In Figure 1I, the cells’ Young’s modulus on the substrate with different stiffness is presented for MCF-7 on the control, Fmoc-FF, and Fmoc-FF+chitosan gels. Chitosan, incorporated into the hydrogel structure, interferes in the peptide self-assembly process, leading to the hydrogels’ softening.

The data obtained in Section 2.1 would be used to assess the adequacy of the alternative SICM method compared to the traditional AFM method for determining the mechanical properties of the cell–soft hydrogel system. The data of AFM analysis would be performed in Section 2.3.

### 2.2. The Correlation of MCF-7 Young’s Modulus Distribution with Fluorescence Imaging

To find out if the obtained Young’s modulus distribution describes the real cell’s structures, the nuclei and microtubules were stained with *Hoechst* 33258 and TubulinTracker^TM^ Green, correspondingly. Figure 2 demonstrates the optical image, topography, Young’s modulus distribution, the fluorescence image of the stained nucleus, and microtubes that were obtained for one cell on control and Fmoc-FF hydrogel.

A comparison of the Young’s modulus distribution for MCF-7 placed on a Petri dish (Figure 2C) and the Fmoc-FF hydrogel (Figure 2I) with the fluorescence images of their nuclei (Figure 2D,J) and tubulin (Figure 2E,K) demonstrates that the stiffest regions on the Young’s modulus distribution map (represented by orange and yellow colors in Figure 2C,I) correspond to areas where the cell nucleus is located, whereas the softer regions (indicated by light blue and blue colors in Figure 2C,I) along the cell periphery correspond to the cytoskeleton.

### 2.3. The Mechanical Properties of the Cell–Hydrogel System Assessed Using AFM

The same systems were examined using the AFM technique. In Figure 3, the optical images (panels A, D and G), topography (panels B, E and H), and Young’s modulus distribution (panels C, F, I) for MCF-7 cells on a Petri dish, Fmoc-FF hydrogel, and Fmoc-FF+chitosan hydrogel are presented.

Due to the high softness of the cells and the gels, and also due to the large cell height, the force maps had artificial stipes along the scanning directions. The affected force curves with a low-quality fit with the Hertz’s model were excluded from the analysis. The typical force curves for all the samples can be seen in Appendix A.

Both optical images show that the shape of the cells changes significantly depending on the substrate: being spread on the stiff Petri dish, they become more rounded on the soft hydrogels. These data are in agreement with the results obtained by the SICM method. In Figure 3J, correspondence between cells’ Young’s modulus and cells’ height on the substrate with different stiffnesses is presented for MCF-7 on the control, Fmoc-FF, and Fmoc-FF+chitosan gels. Chitosan, incorporated into the hydrogel structure, interferes with the peptide self-assembly process, leading to hydrogel’s softening (Figure 3J). Namely, the Young’s modulus for the Fmoc-FF+chitosan system was *E*_mean_ = 450 ± 140 Pa, whereas for the Fmoc-FF gel *E*_mean_ = 500 ± 170 Pa, and the mean height of cells on Fmoc-FF+chitosan exceeded that for Fmoc-FF: *H*_mean (Fmoc+chitosan)_ = 15.2 ± 2.4 μm and *H*_mean (Fmoc)_ = 14 ± 1.5 μm, correspondingly. At that, the mean height of cells on the Petri dish was *H*_mean (Petri dish)_ = 10.9 ± 1.5 μm. These data were taken from the processing of 40 cells for each sample. The schematic illustration of the cell’s shape depending on the substrate stiffness is presented in Figure 3K.

Comparing the Young’s modulus distribution for all the investigated systems, it should be noted that the AFM data are in good agreement with SICM studies: *E*_mean_ = 1190 ± 230 Pa for MCF-7 cells on Petri dish, *E*_mean_ = 980 ± 240 Pa for MCF-7 cells on Fmoc-FF hydrogel, also see the table in the Discussion section. However, the SICM topography and quantitative mechanics mapping had higher resolutions (Figure 1 and Figure 3): for instance, in images acquired using SICM, the distribution of the Young’s modulus of both the cells themselves and the gel is readily discernible. In contrast, the Young’s modulus for the cells and gel obtained through AFM appears similar to each other and it is challenging to differentiate them visually. In AFM experiments, the resolution (80 × 80 points) was limited to improve the data acquisition speed and to avoid excessive sample damage due to repeated indentations. When comparing the topography obtained by both methods, it can be observed that while microvilli with an approximate width of 200–250 nm are visualized using the SICM technique (Figure 1A), they are not discernible in images acquired through the AFM method (Figure 3B). This fact can be explained as follows: the resolution depends on contact radius and the contact radius depends on indentation, which, in turn, depends on sample stiffness. The resolution reduction for AFM data can be explained by higher values of applied forces and higher values of indentation in comparison with the SICM technique: for example, while the mean indentation for the MCF-7+Fmoc-FF hydrogel system was 1500 nm in the AFM experiments, for the same sample, the indentation was evaluated as 220 nm in SICM studies.

### 2.4. The Swelling Determination of Soft Fmoc-FF Hydrogel

The primary research in this article focused on studying the mechanical properties of the cell–soft hydrogel system using the SICM method and comparing the obtained data with those from the traditional AFM method. In order to consider this hydrogel as a potential scaffold in regenerative medicine, its biocompatibility with living cells was examined. When studying the biocompatibility of hydrogels with cells, there is a set of scaffold characteristics that can directly affect cell viability on them. These characteristics include the swelling and degradation of hydrogels. These properties are important because the hydrogel is intended to mimic the extracellular matrix, which is largely composed of water. So, we also conducted swelling studies on Fmoc-FF peptides. The swelling data for Fmoc-FF hydrogels were obtained after incubating the hydrogels in a large volume of PBS (pH 7.4) at 37 °C for 24 h. The results were expressed as the swelling ratio q. It was found that all hydrogels exhibited very high swelling ratios, with q reaching 98%. Sample weight loss measurement data showed that the gels completely degraded within 35 days of incubation, as shown in Appendix A.

### 2.5. Biocompatibility of Soft Hydrogels

After that, the test that describes the biocompatibility of the self-assembling gel and living cells, which is a necessary experiment when proposing a hydrogel as a scaffold, was carried out: MCF-7 cells were placed on the Fmoc-FF hydrogel, which was formed on confocal Petri dishes and washed for 2 days with PBS in order to remove the excess DMSO and to set the hydrogel pH to 7.3 [24]. After that, the samples were UV sterilized. In two days, the live/dead test was carried out using C12 resazurin, a cell membrane dye indicating metabolically active cells (red), and DAPI, a DNA stain indicating dead cells (blue), Appendix A. In addition, a live/dead test demonstrated that there was a dominant population of red cells on Fmoc-FF hydrogel. However, few blue (dead) cells were also found (more than 20 images were studied, each of which contained ~50 cells).

Cytocompatibility was also examined for MCF-7 cells on Fmoc-FF and Fmoc-FF/chitosan hydrogels (Appendix A). For this purpose, DMEM culture medium was incubated with Fmoc-FF and Fmoc-FF+chitosan hydrogels overnight. This medium was then extracted from the hydrogels and added to MCF-7 cells, whose viability was assessed after 24 h of incubation. It can be seen that the presence of chitosan markedly increased cell survival—from 40% up to 80%.

### 2.6. Comparison of Fmoc-FF Hydrogel Biocompatibility Prepared by pH-Switch and Solvent-Switch Methods

Since, in Section 2.5, the low biocompatibility of the Fmoc-FF hydrogel with cells was demonstrated, the question arose whether this result was due to the method of hydrogel preparation or if it was a consequence of the peptide composition of the hydrogel. The self-assembled Fmoc-FF hydrogel can be prepared using two methods: solvent-switch and pH-switch methods. In this work, the hydrogel was prepared by the solvent-switch method: the peptide was dissolved in dimethyl sulfoxide (DMSO) and then diluted in distilled water to a final concentration of *C*_Fmoc-FF_ = 0.6%; the final concentration of DMSO in the solution was 6%. The 6% concentration of DMSO in hydrogel can reduce cell survival [49]. To avoid this effect, the hydrogel was washed for two days with PBS to completely remove excessive DMSO [22]. However, to ensure that it is not excessive DMSO that affects cell survival on hydrogels, a Fmoc-FF gel was prepared using the pH-switch method. The gel was made according to a known procedure [50]: Fmoc-FF peptide was dissolved in water, *C*_Fmoc-FF_ = 0.6%, then NaOH (0.5 M) was added to the solution so that the pH of the solution was adjusted to pH 9, and the solution was stirred using a magnetic stirrer. Thereafter glucono-delta-lactone (GdL) was added to lower the pH to 4, and then the gel was formed and washed with PBS for several hours to adjust pH to 7.3 [24]. Cell biocompatibility tests were performed on Fmoc-FF and Fmoc-FF+chitosan hydrogels (live/dead test and MTT test) that were prepared by pH-switch methods. The results showed that the gel preparation method did not affect the cell survival on the hydrogel. So, we concluded that cell viability was not influenced by the potential residual amount of DMSO in the case of hydrogel preparation by the solvent-switch method.

## 3. Discussion

The main aim of this work was to investigate the mechanical properties of living cell–soft hydrogel systems using SICM, which are able to provide a high resolution, *R* = 40 nm, in aqueous solutions. For the first time, the quantitative nanomechanical distribution of cells–soft hydrogel systems was obtained, see Figure 1D, which enables an estimation of the mechanical properties for both soft biomaterial and the cells placed on it.

The staining of the nucleus and microtubules within cells has enabled a demonstration of the Young’s modulus distribution map, obtained via SICM, reflecting specific cellular structures: namely, the nuclei (the stiffest part of the cells) and the cytoskeleton in the peripheral regions of the cells, see Figure 2. The obtained data can be used to investigate the more detailed response of cells to the influence of the scaffolds on them.

In 2021, the authors Taira et al. [51] presented the topography of endothelial cells placed on collagen hydrogel with nano-resolution obtained using SICM for the first time. However, no quantitative mechanical nano-mapping for the cell–soft hydrogel system was presented. Also, Schierbaum et al. [52] obtained the stiffness of living fibroblasts at a high resolution, *R* = 1 μm, placed on polyacrylamide hydrogel using AFM. At that, it should be noted that the cell was placed on rather rigid hydrogel with a Young’s modulus of *E* = 6.0 ± 0.4 kPa, and the mechanical distribution was represented only for the cell structure without mapping the hydrogel Young’s modulus.

If we compare the values of the Young’s modulus, obtained by AFM, of cells placed on a Petri dish and different hydrogels, we can see the following dependence: the mean *E* for cells were 1190, 980, and 860 Pa placed on the most rigid substrate (Petri dish), Fmoc-FF and Fmoc-FF/chitosan with mean *E* of 500 and 450 Pa. Hence, the less rigid the substrates were, the softer the cells became. The main question was if the cells’ mechanical properties changed due to the cytoskeletal changes in response to their substrates’ stiffness or was it the result of an indentation of substrate that is softer than the cells itself.

Herein, the question of substance softness effect on mechanical measurements should be discussed. The cell on the surface of the hydrogel that is softer than the cell itself can seem softer due to an unaccounted deformation of the underlying substrate during the indentation, which is called the ‘soft substrate effect’. To account for this effect, Rheinlaender et al. suggested a composite cell–substrate model (CoCS) [53], where they represented the total indentation in the experiment as the sum of cell and soft substrate indentations on the example of AFM studies. In this study, a good correlation between the AFM and the SICM methods was demonstrated, as shown in Table 1. So, here, we provide an assessment of how much the cell stiffness might be underestimated in our experiments, using the CoCs model for the spherical indenter using the AFM method and drawing conclusions for both methods given the good agreement of the data between them.

We calculated this underestimation in the cell Young’s modulus obtained by the AFM method, taking into account the typical values of the hydrogel Young’s modulus (450–550 Pa), cell Young’s modulus (800–1000 Pa), the radius of the probe (70 nm), the applied load force (0.1–0.2 nN at 500 nm depth), and the radius of a cell bound to a substrate (10 µm). The obtained estimate was ~7–8%, i.e., relatively small due to a small radius of the used AFM probe, so this correction should be considered for the estimation of cells’ stiffness on soft Fmoc-FF hydrogels. Overall, the softening of the cell on hydrogel was not due to an underestimation of the stiffness of the cell on a soft substrate but was caused by a change in its intrinsic mechanical properties.

Also, the obvious cell shape change was observed in the SICM and AFM experiments during the MCF-7 adhesion on the soft hydrogels in comparison with the stiff Petri dish. The round shape of cells placed on biomaterial versus a spread shape of MCF-7 placed on a Petri dish can be explained by the change in substrate stiffness, see Figure 3J,K, which is in accordance with the investigation of direct influence of matrix stiffness on cell shape [54,55,56]. It was demonstrated that such morphology changes in cells in response to substrate stiffness is due to cytoskeleton variation, specifically to the density and length of the actin fibers. Hogrebe et al. demonstrated that spread cells, placed on a stiff substrate, are characterized by long actin fibers, and their amount increased with the stiffness rising [54]. Moreover, the changes in the substrate stiffness led to different proliferation and differentiation pathways for cells.

Here, it should be noted that not only the substrate’s stiffness but also different functional groups contained on Fmoc-FF hydrogels and on cultured Petri dishes could affect the shape change in MCF-7 cells, which is another factor affecting the topography of cells adsorbed on the substrate [27]. JC Gil-Redondo et al. [57] investigated how the substrate stiffness modulated the viscoelastic properties of MCF-7 cells that were placed on polyacrylamide hydrogels. Polyacrylamide hydrogel substrates with varying Young’s moduli of 0.1 kPa, 4.1 kPa, and 17.3 kPa were considered. The study demonstrated that substrate stiffness had a similar effect on MCF-7 cells as observed in our study, specifically, that with increasing substrate stiffness, MCF-7 cells transitioned from a spherical shape at *E*_substrate_ = 0.1 kPa to a more spread-out form at *E*_substrate_ = 17.3 kPa. This effect was shown using polyacrylamide hydrogel, where the functional groups on the surface remained unchanged with varying stiffness. This could indicate that in our experiment the primary contribution to the change in the shape of MCF-7 cells was due to the alteration in substrate stiffness of Fmoc-FF hydrogels, rather than differences in the functional groups of the gels and the Petri dish.

Another factor that may influence the change in cell shape is their viability on a given substrate. Indeed, cells can change their shape due to the low viability on the Fmoc-FF gel (Appendix A). However, it should be noted that cells on the Fmoc-FF+chitosan gel also exhibit a round shape, even though cell viability in the Fmoc-FF+chitosan gel is high (Appendix A). In the article by JC Gil-Redondo et al. [57], the change in the shape of MCF-7 cells placed on a polyacrylamide gel, which is biocompatible and widely used in regenerative medicine [58], was considered. The study showed that the shape of MCF-7 cells is spherical when they are adsorbed on a soft polyacrylamide gel with a Young’s modulus of 0.1 kPa, whereas increasing the stiffness of the substrate leads to cell spreading. From this, it can be concluded that the stiffness of the hydrogel is the primary factor affecting cell shape.

The biocompatibility analysis for MCF-7 cells on Fmoc-FF hydrogel, *C*_Fmoc-FF_ = 0.6%, demonstrated that Fmoc-FF hydrogel is unsuitable for using this biomaterial as a scaffold. However, the incorporation of chitosan into Fmoc-FF hydrogel, which is known due to its ability to increase biocompatibility and biodegradability [45,46,47,48], increases the cells’ viability from 40% up to 80%. It was demonstrated that the method of hydrogel preparation (solvent switch and pH switch) does not affect cell viability, so a low value of biocompatibility was applied to the chemical structure of hydrogel.

## 4. Materials and Methods

### 4.1. Sample Preparation

#### 4.1.1. Reagents

The *N*-fluorenylmethoxycarbonyl diphenylalanine peptide (Fmoc-Phe-Phe-OH, Fmoc-FF), chitosan (M = 190 kDa), *Hoechst* 33258 were purchased from Sigma-Aldrich (St Louis, MO, USA), *N*-Fmoc-2,3,4,5,6-Pentafluoro-L-phenylalanine (Fmoc-Phe (F5)-OH, Fmoc-F5) was purchased from GL Biochem (Shanghai, China), Dulbecco’s Modified Eagle’s Medium (DMEM) was purchased from Gibco (New York, NY, USA) and TubulinTracker^TM^ Green was purchased from Thermo Fisher Scientific (Waltham, MA, USA).

#### 4.1.2. Hydrogels

The Fmoc-FF hydrogel was prepared using two methods: (a) the solvent-switch method and (b) pH-switch method. The hydrogels using solvent-switch method were prepared as follows: the peptides were dissolved in dimethyl sulfoxide (DMSO) to obtain the stock solution, *C*_Fmoc-FF (stock)_ = 10%, then the stock solution was dissolved in milli-Q water to a final concentration of *C*_Fmoc-FF_ = 0.6% [21,59]. The preparation of hydrogels using the pH-switch method was performed as follows: the weighted peptides were added to milli-Q water, and after that, NaOH solution (0.5 M) was added. The solution was vortexed and sonicated to facilitate mixing until the solution became transparent. The gelation process was initiated by incorporating glucono-d-lactone (GdL) into the basic peptide solution and mixing was accomplished by vortexing for a duration of 5 s. The sample was then left undisturbed. The gelation process took several hours to complete. The solution pH after NaOH addition was 9.1 ± 1.1, the solution pH after GdL addition was 3.9 ± 1.1 [50]. The final concentration of peptide in the hydrogel was *C*_Fmoc-FF_ = 0.6%. After the preparation of hydrogels by both methods, they were carefully washed out with PBS solution for 2 days to ensure complete removal of DMSO [22] (in the case of the solvent-switch method) and also to achieve hydrogel pH 7.3 instead of pH 4 [24]. The Fmoc-FF hydrogels in which the low molecular weight (M = 190 kDa) chitosan was added was prepared as follows: the peptides were dissolved in dimethyl sulfoxide (DMSO) to obtain the stock solution, *C*_Fmoc-FF (stock)_ = 10%; chitosan was dissolved in HCl (1 M), *C*_chitosan (stock)_ = 1%. The chitosan was added into the water to obtain the final concentration 0.6%, *C*_chitosan (final)_ = 0.06%, after that the Fmoc-FF stock solution was mixed with it in such a way to obtain *C*_Fmoc-FF (final)_ = 0.6%, so *C*_Fmoc-FF_:*C*_chitosan_ = 10:1. The hydrogel was washed out for several hours with PBS buffer to obtain the final pH of hydrogel 7.3.

Chemical structure (Appendix A) and Fourier Transform InfraRed (FTIR) analysis (Appendix A) are inserted in Appendix A for thorough characterization of the investigated hydrogel.

#### 4.1.3. Cell Culturing

MCF-7 cell lines were grown in DMEM F/12 complete culture medium (PanEco, Moscow, Russia) containing 10% fetal calf serum (FBS) (Gibco, New York, NY, USA), 1x PenStrep (Gibco, New York, NY, USA) and 1x GlutaMax (Gibco, New York, NY, USA) in a standard culture incubator containing 5% CO_2_ and maintained at 37 °C. For staining and subsequent CIPM assay, cells were seeded on gels formed on confocal cups at an amount of 300,000 per cup and incubated at 37 °C, 5% CO_2_ for 24 h. Cells seeded on confocal cups without gels were used as control.

For cytotoxicity test, cells were seeded into the wells of 96-well plates 24 h before the experiment in the amount of 12 thousand cells per well. After the incubation period, eluted media were added to the wells with cells and the cells were incubated at 37 °C, 5% CO_2_ for a day. As a control, wells with cells in which the medium was changed normally were taken.

### 4.2. Scanning Ion Conductance Microscopy (SICM) Measurements

The SICM method (ICAPPIC Limited, London, UK) was employed to study the topography and mechanical properties of the hydrogel. The Olympus IX73 inverted optical microscope (Tokyo, Japan) was utilized in the experimental processes. The ICAPPIC Universal Controller and Piezo Control System (ICAPPIC Limited, London, UK) managed the feedback control and piezo positioning. A P-2000 laser puller (Sutter Instruments, San Rafael, CA, USA) produced borosilicate glass pipettes for topography analysis. The exact radius of capillary *R_cap_* value was determined using a formula (1) [60]:
*I* = *πR_cap_kVtan*(*α*)(1)


Here, *I* signifies ion current, *α* denotes a half-cone angle, which was equal to 3 degrees, *k* is 1.35 Sm^−1^, and *V* represents the applied voltage, which was 200 mV in the experiment. The hydrogel’s topography was examined using the noncontact hopping mode, with a decrease in ion current of 0.5% and a scan size of 20 × 20 µm or 25 × 25 µm. The final image resolution was 256 × 256 pixels. The fall rate during topography studies was set at 120 µm·s^−1^.

For the hydrogel Young’s modulus values the empirical formula was applied (2) [41]:(2)$$E=pA{\left(\frac{S_{sub}}{S_{samp}}-1\right)}^{-1}=pA{\left(\frac{∆I/d_{sub}}{{∆I/d}_{samp}}-1\right)}^{-1}=pA{\left(\frac{d_{samp}}{d_{sub}}-1\right)}^{-1}$$
where *E* represents Young’s modulus and *A* signifies a constant reliant on the pipette’s geometric attributes. The geometrical parameter *A* for different values of inner half cone angle a and wall thickness of the pipette is presented in Appendix A in [41]. The parameters of applied radii *R*, ion currents *I*, and colloidal pressures can be seen in Appendix A.

The pressure applied is denoted as *p*, while *S_sub_* and *S_samp_* refer to the current–distance curve slopes for a 1% and 2% decrease in ion current for a substrate and a sample, respectively. The current–distance curve slopes are defined by current drop of 1% and 2% and indentations of the substrate and the sample *d_sub_* and *d_samp_*. Since current drop is the same for both the substrate and the sample, to obtain the mechanics map, it is necessary to record the deformation on the substrate and the sample at a current drop of 1% and 2% and applied pressures.

In our study, we utilized equation (2) to determine the pressure *p*, colloidal pressure, which was derived from the DLVO theory, accounting for the inherent colloidal pressure that develops within the interfacial liquid layers as they become significantly compressed when the pipette is brought near the sample. It should be noted that usually the Debye length in PBS solution is considered as 1 nm. However, this is correct if only electrostatic interactions of double layers are considered. Clarke et.al. clearly explained that interlayers did not only contribute to electrostatic forces at separation of the size and magnitude of the Hamaker constant across gaps of tens of nanometers because of the drastic variations in permittivity across these interlayers [61].

In the experiment, the pressure *p* was estimated based on the intrinsic force as detailed by Kolmogorov et al. [36], where it was calculated on the example of decan drop. The derived formula for the intrinsic force *F* was expressed as *F =* 4π*σ*(*R_cap_*·*d*)^1/2^, where σ represents the surface tension coefficient, *R*—radius of capillary, and *d*—indentation. The colloidal pressure was determined as *p = F*/*S = F*/π*a*^2^, where *a* is a contact radius, *a = (R_cap_·d*)^1/2^. A detailed explanation of the force and pressure calculations based on the experimental parameters is provided in the Appendix A.

When assessing the mean value *E* for cells on hydrogels, more than 40 cells were analyzed (both for cells on Fmoc-FF hydrogel and for cells on Fmoc-FF+chitosan hydrogel). To determine this parameter with the highest accuracy, individual cells or hydrogels were separately identified in each frame, and the mean value *E* was calculated for these objects.

### 4.3. Fluorescence Study of Stained Cells. Confocal Measurements

In order to obtain the fluorescence image of stained cell nuclei and microtubes, which consist of tubulin, the light from the light-emitting diode (LED) was focused on samples with the use of DAPI filter set 49 (λ_ex_ = 365 nm, λ_em_ = 445 nm) and GFP filter set 38HE eGFP (λ_ex_ = 470, λ_em_ = 525 nm). The experimental setup can be seen in Appendix A.

For the visualization of microtubules within MCF-7 cells via fluorescence studies, the cells were labeled with TubulinTracker^TM^ Green at a final concentration of 1 μg/mL. Cells were incubated with the dye at 37 °C, 5% CO_2_ for 1 h in DMEM/F-12 complete nutrient medium. Cells were also stained with Hoechst dye at a final concentration of 5 μg/mL (8.115 μM) to visualize cells nuclei. Hoechst was added to cells 20 min before the end of incubation with Tubulin Tracker. After the end of incubation time, cells were washed with HBSS 1–2 times.

### 4.4. Atomic Force Microscopy Measurements

The atomic force microscopy (AFM) method was used for the assessment of cell–hydrogel system stiffness using a Bruker Bioscope Resolve AFM (Bruker, Billerica, MA, USA). For these tests, the implementation of force volume mode to determine mechanical attributes was applied. The PeakForce QNM-Live Cell probes (PFQNM-LC-A-CAL, Bruker AFM Probes, Camarillo, CA, USA) with a pre-calibrated spring constant (around 0.1 N/m) and a parabolic tip with a radius of 70 nm (for estimation of tip shape see Appendix A) were used. Images of 80 × 80 µm^2^ were obtained at a resolution ranging from 40 × 40 to 80 × 80 pixels. Force curves were acquired at a piezo speed of 180 μm·s^−1^, with a force set-point between 0.5–1 nN. The indentation depth varied from 500 to 1500 nm in the tests depending on the stiffness of the sample.

Hertz’s model (3) was used for the numerical processing of force curves using the MATLAB routine (The MathWorks, Natick, MA, USA) [62]:*F*(*δ*) = 4*R_tip_*^1/2^/(3·(1 − *ν*^2^))*Eδ*^3/2^(3)

In this model, *F* represents the force acting on the cantilever tip, *δ* is the indentation depth, *R_tip_* stands for the indenter radius, *ν* is the Poisson’s ratio of the sample (assumed to be 0.5 for the cells [63,64] and hydrogel [65] as it was shown that they maintain their volume under load), and *E* is the Young’s modulus. The model characterizes the indentation of a parabolic tip into an elastic half-space and is applicable to indentation depths exceeding the tip radius [66]. Here, the indentation range for the fit was limited to 500 nm. In AFM experiments, to determine the Young’s modulus, the fit was applied only to the initial 500 nm of indentation depth for investigated systems. This range fell within the applicability range of the Hertz model for the AFM probes with parabolic tips used, as demonstrated in previous studies [66,67]. While the Hertz model can be used for spherical indenters at depths below the sphere’s radius, the parabolic tip shape does not have this limitation [68]. Additionally, the tip shape was analyzed by scanning a calibration grating and was well-fitted with a quadratic function up to 500 nm of tip height. The estimated radius of curvature at the peak matched well with the tip radius provided by the probe manufacturer (70 nm ± 10%). The chosen depth of 500 nm allows for the analysis of samples within the same surface layer, independent of local stiffness and the highest applied load. Limiting the indentation depth also helps to remain within the linear elastic regime of the material [69]. For the tips used, a transition from a paraboloid to a 3-sided pyramid occurs after an indentation of around 550 nm [66].

The regions with cells and bare gel surface were selected on the force volume maps. Force curves with low quality of fit were excluded from the analysis. More than 40 cells and 15 bare gel areas were analyzed per sample.

### 4.5. Swelling and Degradation of Fmoc-FF Hydrogel

To acquire insights into the physical structure of Fmoc-FF hydrogels, we assessed the swelling ratio for samples. Swelling ratios for Fmoc-FF hydrogels with identical initial total concentrations were determined using PBS. Initially, all Fmoc peptide-based hydrogels were weighed (Wi). Subsequently, 3 mL of PBS (pH 7.4) were added to each hydrogel, which were then incubated at 37 °C overnight. The fully swollen hydrogels were weighed (Ws) immediately after the removal of excess water. Following this, the hydrogels were freeze-dried and weighed again (Wd). The swelling behavior was quantified as the swelling ratio q (%), defined as the ratio of the swollen sample weight (Ws) to the freeze-dried hydrogel weight (Wd) (q = (Ws − Wd)/Ws·100). The mass loss was determined by the formula ΔW = Ws/Wi.

### 4.6. MTT Test

For the MTT test, gels were prepared in a 96-well plate under sterile conditions and washed 5 times with DPBS overnight. Then, the gels were incubated at 37 °C, 5% CO_2_ with complete culture medium overnight. The eluted medium was transferred to cells in a 96-well plate. The cells were incubated with transferred medium for 24 h, after which the MTT test was carried out: the medium with MTT reagent was added, the cells were incubated for 4 h, then the medium with reagent was withdrawn, DMSO was added and cells were incubated on a shaker for 15 min. Fluorescence intensity was measured using a multi-plate reader (Varioskan LUX, Thermo Fisher Scientific Inc., Waltham, MA, USA).

### 4.7. Live/Dead Test

The percentage of surviving cells on the gels was determined by staining them with two fluorescent dyes: C12 Resazurin—red dye, which stains the membrane of metabolically active cells, and DAPI—blue dye, which stains DNA in cells with damaged membranes (late apoptotic and necrotic).

Gels were prepared under sterile conditions in confocal Petri dishes, then washed 5 times with DPBS for a day. Afterward, cells were seeded on the dishes at a rate of 300,000/dish. The cells were incubated at 37 °C, 5% CO_2_ for 48 h. At the end of incubation period, cells were stained: DAPI (*C*_DAPI_ = 1mM) and C12 Resazurin (*C*_res_ = 0.5mM) were added; incubation with dyes was carried out at 37 °C, 5% CO_2_ for 10 min. The live/dead test measurements were carried out using inverted fluorescence microscope (EVOS M5000 Imaging System, Thermo Fisher Scientific Inc., Waltham, MA, USA).

## 5. Conclusions

The aim of this article was to explore what new possibilities or information the alternative scanning ion conductance microscopy (SICM) method can provide compared with the traditional atomic force microscopy (AFM) method in studying the mechanical properties of the cell–hydrogel system. To the best of our knowledge, there are no studies that have produced maps of the mechanical property distribution of cell–hydrogel systems where the substrate is softer than the cell itself. Such systems might be of interest in the field of regenerative medicine for the restoration of soft tissues, such as soft brain tissue, E ~ 0.1–1 kPa [10], or adipose tissues E ~ 1–2 kPa [11]. The use of this method allowed us, firstly, to simultaneously obtain a distribution map of both the soft hydrogel and the living cell. Such measuring gives us the opportunity to examine the immediate changes in cells’ mechanical properties in response to scaffold behavior, which is important in the field of regenerative medicine when testing the effects of different drugs on the cell–scaffold system. Secondly, the measurements carried out with SICM were in agreement with the AFM technique. One of the main advantages of SICM over AFM is its non-invasiveness; therefore, the data obtained in this study suggest that the SICM method can be used as an independent method for measuring the mechanical properties of the cell–soft hydrogel system. Thanks to its non-destructive nature, it allows for the investigation of the dynamics of the same cell on the hydrogel over a long period, which cannot be achieved with the traditional AFM method. This capability is also important for obtaining more accurate information about changes in cell metabolism during their interaction with the scaffold. Furthermore, studies using confocal microscopy showed that the mechanical property distribution map of the cell on the hydrogel obtained by the SICM method describes specific cellular structures (nucleus/cytoskeleton), allowing for a more detailed description of changes in cell biomechanics during interaction with the hydrogel.

The biocompatibility tests demonstrated the poor cell viability of the Fmoc-FF hydrogel, though the incorporation of chitosan into self-assembled hydrogel improved the obtained value. In the future, it is planned to modify the composition of the gel to make it more suitable as a biocompatible scaffold for living cells. This could be achieved, for example, by adding other peptides such as Fmoc-F5 or PEG8-(FY)3 hexapeptide [22]. Subsequently, specific tissue types can be targeted by using epithelial or neuronal cells (depending on the task at hand). This will allow for the dynamic study of changes in cell metabolism when adhered to the surface of self-assembling gels, the effects of growth factors and other substances on the systems under investigation, and simultaneously, the degradation of the gel itself using the SICM method. The carried-out procedure can be adapted for the investigation of any cell–soft hydrogel system in order to identify the most biocompatible soft hydrogel for application in regenerative medicine.

## Figures and Tables

**Figure 1 ijms-25-13479-f001:**
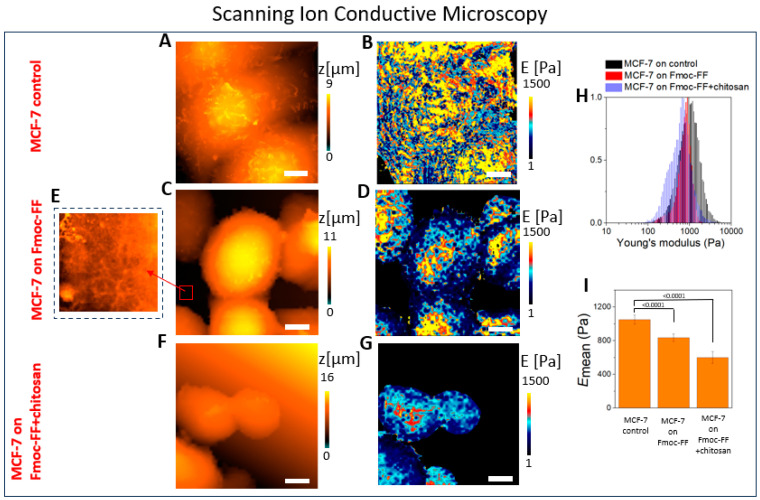
The SICM topography image of MCF-7 cells placed on (**A**) a cultural Petri dish, (**C**) Fmoc-FF hydrogel, and (**F**) Fmoc-FF+chitosan hydrogel. The distribution of Young’s modulus for MCF-7 cells placed on (**B**) cultural Petri dish, (**D**) Fmoc-FF hydrogel, and (**G**) Fmoc-FF+chitosan hydrogel, *C*_Fmoc-FF_ = 0.6%, *C*_Fmoc-FF_:*C*_chitosan_ = 10:1. (**E**) The SICM topography image of Fmoc-FF hydrogel. (**H**) The histogram of Young’s modulus distribution for MCF-7 cells placed on cultural Petri dish (control, black lines), Fmoc-FF hydrogel (red lines), and Fmoc-FF+chitosan hydrogel (violet lines). (**I**) The cell Young’s modulus for MCF-7 on control, Fmoc-FF hydrogel, and Fmoc-FF+chitosan hydrogel systems. The scale bar is 5 μm. The radii of capillaries were *R_cap_* = 45 nm.

**Figure 2 ijms-25-13479-f002:**
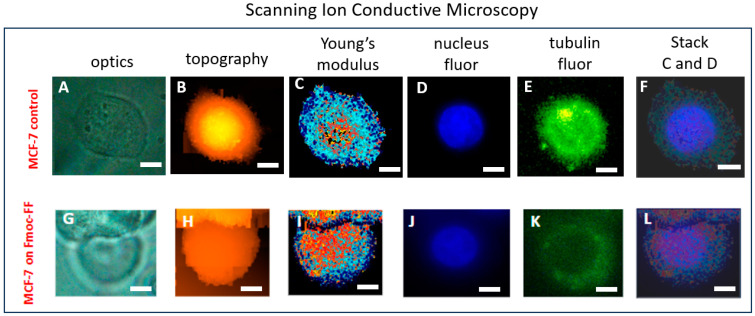
(**A**,**G**) optical images, (**B**,**H**) topography, (**C**,**I**) Young’s modulus distribution, (**D**,**J**) fluorescence image of nuclei, (**E**,**K**) fluorescence image of tubulin, (**F**,**L**) stack images of Young’s modulus distribution and fluorescence image of nuclei for MCF-7 cells placed on cultural Petri dish and Fmoc-FF hydrogel, correspondingly. The scale bar is 5 μm.

**Figure 3 ijms-25-13479-f003:**
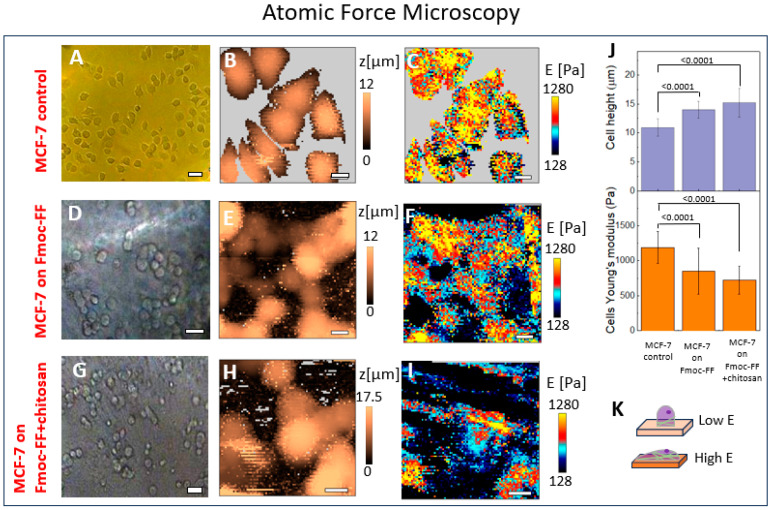
The optical images (**A**,**D**,**G**), topography (**B**,**E**,**H**), and Young’s modulus distribution (**C**,**F**,**I**) for MCF-7 cells on Petri dish, Fmoc-FF hydrogel, and Fmoc-FF/chitosan hydrogel, correspondingly. (**J**) The cell height and cell Young’s modulus for MCF-7 on control, Fmoc-FF hydrogel, and Fmoc-FF+chitosan hydrogel systems. (**K**) The schematic images for cells placed on low and high modulus substances. The scale bar for optical images is 40 μm; the scale bar for topography and Young’s modulus images are 10 μm. *C*_Fmoc-FF_ = 0.6%, *C*_Fmoc-FF_:*C*_chitosan_ = 10:1. Statistical significance was probed with Kruskal–Wallis non-parametric test and Dunn’s correction for multiple comparisons. *R_tip_* = 70 nm.

**Table 1 ijms-25-13479-t001:** The mean Young’s modulus for 3 systems: MCF-7 on Petri dish (control), MCF-7 on Fmoc-FF hydrogel, MCF-7 on Fmoc-FF/chitosan hydrogel. *C*_Fmoc-FF_ = 0.6%, *C*_Fmoc-FF_:*C*_chitosan_ = 1:1. The data were obtained by SICM and AFM methods.

		SICM	AFM	AFM, CoCS Model
MCF-7 on Petri dish, *E*, Pa		1050 ± 55	1190 ± 230	-
MCF-7 on Fmoc-FF hydrogel, *E*, Pa	MCF-7	835 ± 45	980 ± 240	1060 ± 250
	Fmoc-FF	300 ± 30	500 ± 170	-
MCF-7 on Fmoc-FF+chitosan hydrogel, *E*, Pa	MCF-7	600 ± 70	860 ± 220	930 ± 230
	Fmoc-FF+chitosan	270 ± 60	450 ± 140	-

## Data Availability

The authors confirm that the data supporting the findings of this study are available within the article and its Appendix A.

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
