# Peer review of "Non-Invasive Nanometer Resolution Assessment of Cell–Soft Hydrogel System Mechanical Properties by Scanning Ion Conductance Microscopy"

_ijms, 2024, doi:10.3390/ijms252413479_

Round 1

Reviewer 1 Report

Comments and Suggestions for Authors

Major revision:

1. The authors evaluated the mechanical properties of the cell-soft-hydrogel system at non-invasive nanoscale resolution using scanning ion conduction microscopy. In simple terms, the authors should just use the microscope to show the mechanical properties of cells in different states (culture dish and hydrogel), so what is the significance? What can we do with all this data? Can it be used as an indicator of key behaviors? The authors are invited to explain in detail.

2. In Section 2.1, using SICM to evaluate the mechanical properties of the cell-hydrogel system, the authors only describe the relevant parameters obtained without specific explanation. What is the use of these data after we obtain them? Can these data be used to evaluate the hydrogel model? Or is there some other use? And then there's Section 2.2.

3. It can be seen from the title that the author mainly uses some technical means to evaluate the resolution of hydrogels. What is the purpose of the tests in parts 2.4, 2.5 and 2.6? What is the relevance to the subject? These are some of the common reported preparation methods and related properties of hydrogels. Please explain in detail.

Minor revision:

1. The way in which the author uses the number of figures in the text is inconsistent, such as fig.xx, Figure xx, Fig.XX, Fig Xx, Fig xx(A), Fig xxA. Some lack space. This problem appears many times in the main text, please correct it.

2. In Figure 1, there is no G, I in the number. The numbering order and the back and forth order of the text description are also inconsistent. The numbers of B and D are in the interior of the figure and are not clear. Suggestions are written outside the picture. You inserted a picture in C, and I'm sure you can add a number as well. In J, the author describes its bimodal nature, but it is not very clear, which may be overlap and color occlusion. It is suggested to divide it into three separate figures in SI.

3. Figure 2. The size of the picture is inconsistent, and the scale bar does not appear in the explanation.

Author Response

We thank the reviewer for the positive assessment of the work and appreciate the time and effort invested.

All the corrections that were added into the manuscript according to the Reviewer suggesting were highlighted with yellow.

Major revision:

Comments 1: The authors evaluated the mechanical properties of the cell-soft-hydrogel system at non-invasive nanoscale resolution using scanning ion conduction microscopy. In simple terms, the authors should just use the microscope to show the mechanical properties of cells in different states (culture dish and hydrogel), so what is the significance? What can we do with all this data? Can it be used as an indicator of key behaviors? The authors are invited to explain in detail.

Response 1:

The aim of this article was not to obtain indicators of the key behavioral parameters of MCF-7 cells adhered to self-assembling peptide hydrogels, but rather to explore what new possibilities or information the alternative Scanning Ion-Conductance Microscopy (SICM) method can provide compared to the traditional Atomic Force Microscopy (AFM) method in studying the mechanical properties of the cell-hydrogel system. To the best of our knowledge, there are no studies that have produced maps of the mechanical property distribution of cell-hydrogel systems where the substrate is softer than the cell itself. Such systems might be of interest in the field of regenerative medicine for the restoration of soft tissues, such as soft brain tissue, E ~ 0.1-1 kPa, or adipose tissues E ~ 1-2 kPa. The use of this method allowed us, firstly, to simultaneously obtain a distribution map of both the soft hydrogel and the living cell. Such measuring gives the opportunity to examine the immediate change of cell’s mechanical properties in response to scaffold behavior, which is important in the field of regenerative medicine when testing the effects of different drugs on the cell-scaffold system. Secondly, the measurements carried out with SICM were in agreement with the AFM technique. One of the main advantages of SICM over AFM is its non-invasiveness; therefore, the data obtained in this study suggest that the SICM method can be used as an independent method for measuring the mechanical properties of the cell-soft hydrogel system. Thanks to its non-destructive nature, it allows for the investigation of the dynamics of the same cell on the hydrogel over a long period, which cannot be achieved with the traditional AFM method. This capability is also important for obtaining more accurate information about changes in cell metabolism during their interaction with the scaffold. Furthermore, studies using confocal microscopy showed that the mechanical property distribution map of the cell on the hydrogel obtained by the SICM method describes specific cellular structures (nucleus/cytoskeleton), allowing for a more detailed description of changes in cell biomechanics during interaction with the hydrogel.

These considerations were added to the Conclusions section.

Comments 2: In Section 2.1, using SICM to evaluate the mechanical properties of the cell-hydrogel system, the authors only describe the relevant parameters obtained without specific explanation. What is the use of these data after we obtain them? Can these data be used to evaluate the hydrogel model? Or is there some other use? And then there's Section 2.2.

Response 2:

The data obtained in Section 2.1 were used to assess the adequacy of the alternative SIСM method compared to the traditional AFM method for determining the mechanical properties of the cell-soft hydrogel system. By comparing the data from Section 2.1 and Section 2.3 in the Discussion section, it is concluded that this method correlates well with the AFM method, which allows for the advantages of the SIСM method over the AFM method to be utilized in the study of the biomechanics of live cells adhered to scaffolds. The additional information obtained using the SIСM method compared to AFM and its potential applications are described in the Conclusion section.

Section 2.2, as mentioned at the beginning of the section, is dedicated to investigating whether the mechanical property distribution maps obtained using the SICM method correspond to real organelles in cells. The obtained data can be used to investigate the more detailed response of cells to the influence of scaffolds on them.

Explanatory information has been added at the end of Sections 2.1, Discussion and Conclusion sections.

Comments 3: It can be seen from the title that the author mainly uses some technical means to evaluate the resolution of hydrogels. What is the purpose of the tests in parts 2.4, 2.5 and 2.6? What is the relevance to the subject? These are some of the common reported preparation methods and related properties of hydrogels. Please explain in detail.

Response 3:

The primary research in this article focused on studying the mechanical properties of the cell-soft hydrogel system using the SICM method and comparing the obtained data with those from the traditional AFM method. However, a logical continuation of this research is the study of the self-assembling hydrogel to investigate its biocompatibility with living cells, with the aim of using it as a scaffold in regenerative medicine in the future.

When studying the biocompatibility of hydrogels with cells, there is a set of scaffold characteristics that can directly affect cell viability on them. These characteristics include the swelling and degradation of hydrogels. These properties are important because the hydrogel is intended to mimic the extracellular matrix, which is largely composed of water.

Section 2.5 is dedicated to the test that describes the biocompatibility of the self-assembling gel and living cells, which is a necessary experiment when proposing a hydrogel as a scaffold.

Since section 2.5 demonstrated low biocompatibility of the Fmoc-FF hydrogel with cells, the question arose whether this result is due to the method of hydrogel preparation (which could be adjusted to improve the biocompatibility of the gel and cells) or if it is a consequence of the peptide composition of the hydrogel. This was investigated in this section.

The explanations have been added to sections 2.4-2.6.

Minor revision:

Comments 1: The way in which the author uses the number of figures in the text is inconsistent, such as fig.xx, Figure xx, Fig.XX, Fig Xx, Fig xx(A), Fig xxA. Some lack space. This problem appears many times in the main text, please correct it.

Response 1:

This problem was corrected.

Comments 2: In Figure 1, there is no G, I in the number. The numbering order and the back and forth order of the text description are also inconsistent. The numbers of B and D are in the interior of the figure and are not clear. Suggestions are written outside the picture. You inserted a picture in C, and I'm sure you can add a number as well. In J, the author describes its bimodal nature, but it is not very clear, which may be overlap and color occlusion. It is suggested to divide it into three separate figures in SI.

Response 2:

Figure 1 and its numbering order was corrected. To clarify the bimodal nature of system the Fig. 1H was divided into three separate figures and also presented in SI, see Fig. S3.

Comments 3: Figure 2. The size of the picture is inconsistent, and the scale bar does not appear in the explanation.

Response 3: The scale bar was added to Figure 2 and to the explanation of this Figure.

Reviewer 2 Report

Comments and Suggestions for Authors

The manuscript titled “Non-invasive nanometer resolution assessment of cell-soft-hydrogel system mechanical properties by scanning ion-conductance microscopy” by Tikhonova, T.N.; et al. is a scientific work where the authors combine scanning ion conductance microscopy (SICM) and atomic force microscopy (AFM) to extract the elastic modulus of carcinogenic MCF-7 breast cancer cells seeded on substrates with different stiffness. This study could be interesting to optimize and standardize protocols concerning how the substract can impacts on the Young’s modulus of the examined cellular samples. The manuscript is generally well-written and this is a topic of growing interest.

However, it exists some points that need to be addressed (please, see them below detailed point-by-point) to improve the scientific quality of the submitted manuscript paper before this article will be consider for its publication in the International Journal of Molecular Sciences.

1) “(…) ‘soft substrate (…) (lines 33-34). The authors need to fix this grammar issue.

2) Keywords. The authors should consider to add the term “MCF-7 breast cancer cells” in the keyword list.

3) “Hydrogels can be broadly categorized into shynthetic and natural materials. Syntheti hydrogels, engineered from polymers (…) crosslinking density” (lines 52-54). Could the authors provide some illustrative examples?

4) Then, during the Introduction section it is also neccesary to discuss about the existence of other alternative techniques to ascertain the mechanical properties of the examined sample. In this framework, optical tweezers [1] and magnetic resonance elastography [2] need to be mentioned highlighting how the combination of AFM and SICM offer advantages compared to the above described tools.

[1] Magazzù, A.; et al. Investigation of Soft Matter Nanomechanics by Atomic Force Microscopy and Optical Tweezers: A Comprehensive Review. Nanomaterials 2023, 13, 963. https://doi.org/10.3390/nano13060963

[2] Kim, S.H.; et al. Magnetic Resonance Elastography for the Detection and Classification of Prostate Cancer. Cancers 2024, 16, 3494. https://doi.org/10.3390/cancers16203494

5) Finally, could the authors provide quantitative data insights according to the worldwide global incidence burdens of breast cancer malignancies? This will significantly aid the potential readers to better understand the significance of this research.

6) This reasearch assessed the mechanical properties of MCF-7 malignant breast cancer cells but it need also to test benign breast cells as negative control (e.g. MCF10A breast cell lines).

7) Figure 2 (line 155). The lateral scale bar needs to be furnish for the images displayed in all the panels.

8) “The resolution reduction of AFM data (…) higher values of applied forces and higher values of indentation (…) 1500 nm in AFM experiments (…) 220 nm in SICM studies” (liens 211-215). The elastic modulus obtention directly relies on the exerted load forces and the indentation depths. These values are required to be consistent among AFM ans SICM experiments in order to make comparable the data gathered by both techniques.

9) Materials & Methods. “Chemical structure (Fig. S1 (A)) and FTIR analysis (…)” (line 385). The full-name needs to be stated the first time that a term appears in the text. Then, the abbreviation should be placed between brackets. This comment should be taken into account for the rest of the main manuscript body text.

10) “(…) υ is the Poisson ration of the sample (assumed to be 0,5 for the cells and hydrogel)”. (lines 473-474). Why did the authors select a Poisson ratio of 0.5 instead of lower values (e.g. 0.30 or 0.35)? A brief explanation should be furnished in this regard.

11) “5. Conclusions” (lines 525-537). This section perfectly remarks the most relevant outcomes found by the authors in this work and also the promising future prospectives. The authors should furnish a brief statement to discuss about the future action lines to pursue the topic covered in this work.

Author Response

Overview comment: The manuscript titled “Non-invasive nanometer resolution assessment of cell-soft-hydrogel system mechanical properties by scanning ion-conductance microscopy” by Tikhonova, T.N.; et al. is a scientific work where the authors combine scanning ion conductance microscopy (SICM) and atomic force microscopy (AFM) to extract the elastic modulus of carcinogenic MCF-7 breast cancer cells seeded on substrates with different stiffness. This study could be interesting to optimize and standardize protocols concerning how the substract can impacts on the Young’s modulus of the examined cellular samples. The manuscript is generally well-written and this is a topic of growing interest.

Response: We thank the reviewer for the positive assessment of the work and appreciate the time and effort invested.

All the corrections that were added into the manuscript according to the Reviewer suggesting were highlighted with yellow.

Comments 1: “(…) ‘soft substrate (…) (lines 33-34). The authors need to fix this grammar issue.

Response 1: The term "soft substrate effect" was taken from the composite model proposed by Rheinlaender et al. [ Rheinlaender 2020], which was developed to account for the additional deformation of the substrate that may occur in a cell-soft substrate system, where the substrate is softer than the cell. This model is described in Discussion section and used for AFM data processing.

- Rheinlaender, J.; Dimitracopoulos, A.; Wallmeyer, B.; Kronenberg, N.M.; Chalut, K.J.; Gather, M.C.; Betz, T.; Charras, G.; Franze, K. Cortical cell stiffness is independent of substrate mechanics. Nat. Mater. 2020, 19, 1019-1025.

Comments 2: Keywords. The authors should consider to add the term “MCF-7 breast cancer cells” in the keyword list.

Response 2: The term “MCF-7 breast cancer cells” was added to the keyword list.

Comments 3: “Hydrogels can be broadly categorized into synthetic and natural materials. Synthetic hydrogels, engineered from polymers (…) crosslinking density” (lines 52-54). Could the authors provide some illustrative examples?

Response 3: The illustrative examples of synthetic hydrogels were added into the sentence: “Synthetic hydrogels, engineered from polymers such as polyethene glycol and polyamides offer the advantage of tunability, where their mechanical properties can be precisely controlled by altering the crosslinking density [15].”

Comments 4: Then, during the Introduction section it is also necessary to discuss about the existence of other alternative techniques to ascertain the mechanical properties of the examined sample. In this framework, optical tweezers [1] and magnetic resonance elastography [2] need to be mentioned highlighting how the combination of AFM and SICM offer advantages compared to the above described tools.

 [1] Magazzù, A.; et al. Investigation of Soft Matter Nanomechanics by Atomic Force Microscopy and Optical Tweezers: A Comprehensive Review. Nanomaterials 202313, 963. https://doi.org/10.3390/nano13060963

[2] Kim, S.H.; et al. Magnetic Resonance Elastography for the Detection and Classification of Prostate Cancer. Cancers 202416, 3494. https://doi.org/10.3390/cancers16203494

Response 4:

In recent years, the exploration of mechanical properties of materials at the nanoscale has garnered significant attention, leading to the development and application of various techniques in addition to the traditional, well-recognized AFM method. Among these, optical tweezers, as reviewed by Magazzù et al. [Magazzu], utilize highly focused laser beams to manipulate and measure the mechanical properties of soft matter at the nanoscale. For instance, optical tweezers have been used to study changes in the characteristic elasticity of cells associated with various human diseases [Dao 2003, Agrawal 2016]. This technique is particularly advantageous for its non-invasive nature and high precision; however, it is effective only for manipulating very small objects, such as microparticles and molecules, which limits its application to larger samples. Additionally, the method's effectiveness decreases when working with thicker or opaque samples. Methods are also being developed that can be used directly in clinical settings to determine tissue stiffness in patients with diseases such as prostate cancer [Kim 2024]. One such method is magnetic resonance elastography. Although this technique offers a non-invasive imaging method that measures the mechanical properties of tissues, making it valuable for medical applications, it has lower spatial resolution compared to methods like optical tweezers or AFM, which may limit its application for studying small structures.

- Magazzù, A.; et al. Investigation of Soft Matter Nanomechanics by Atomic Force Microscopy and Optical Tweezers: A Comprehensive Review. Nanomaterials 202313, 963. https://doi.org/10.3390/nano13060963

-Dao, M.; Lim, C.T.; Suresh, S. Mechanics of the human red blood cell deformed by optical tweezers. J. Mech. Phys. Solids 200351, 2259–2280.

-Agrawal, R.; Smart, T.; Nobre-Cardoso, J.; Richards, C.; Bhatnagar, R.; Tufail, A.; Shima, D.; Jones, P.H.; Pavesio, C. Assessment of red blood cell deformability in type 2 diabetes mellitus and diabetic retinopathy by dual optical tweezers stretching technique. Sci. Rep. 20166, 15873.

- Kim, S.H.; et al. Magnetic Resonance Elastography for the Detection and Classification of Prostate Cancer. Cancers 202416, 3494. https://doi.org/10.3390/cancers16203494

Comments 5: Finally, could the authors provide quantitative data insights according to the worldwide global incidence burdens of breast cancer malignancies? This will significantly aid the potential readers to better understand the significance of this research.

Response 5:

In this study, MCF-7 breast cancer cells were chosen as a convenient and well-researched model for investigating the mechanical properties of the cell-soft hydrogel system using a novel alternative method, SIСM. For instance, in study [39], the morphology of MCF-7 cell membranes was examined in relation to substrate stiffness, specifically changes in the number of microvesicles, microvilli, and the structure of filopodia. Gil-Redondo et al. demonstrated changes in the shape of these cells depending on the substrate stiffness [53]. It was hypothesized that if this convenient model could show that the SIСM method allows for a deep characterization of the cell-soft hydrogel system, with the substrate stiffness being lower than that of the cells, and that the self-assembling Fmoc-FF hydrogel would be a promising biocompatible scaffold for these cells, then this method could be used in the future to study various epithelial/neural or other cell types. This includes investigating their mechanical properties after treatment with drugs, growth factors, or hydrogel degradation. These considerations were added to the Introduction section.

Comments 6: This research assessed the mechanical properties of MCF-7 malignant breast cancer cells but it need also to test benign breast cells as negative control (e.g. MCF10A breast cell lines).

Response 6:

As previously noted, MCF-7 breast cancer cells were used in this study not for the investigation of breast cancer malignancies, but as a convenient and well-researched model. Their biocompatibility with the self-assembling Fmoc-FF hydrogel can be considered a reference point, against which the potential use of this gel with other, more demanding cell types can be evaluated. These considerations were added to the Introduction section.

Comments 7: Figure 2 (line 155). The lateral scale bar needs to be furnish for the images displayed in all the panels.

Response 7:

The scale bar was added to all the images in Figure 2.

Comments 8: “The resolution reduction of AFM data (…) higher values of applied forces and higher values of indentation (…) 1500 nm in AFM experiments (…) 220 nm in SICM studies” (liens 211-215). The elastic modulus obtention directly relies on the exerted load forces and the indentation depths. These values are required to be consistent among AFM ans SICM experiments in order to make comparable the data gathered by both techniques.

Response 8:

One of the main advantages of SICM over AFM is its noninvasiveness, i.e., the absence of direct capillary impact on cells. Therefore, using the same loading force and indentation depth for both experiments would make it pointless to use the SICM method to study the living cell-hydrogel system. 

On the other hand, unfortunately, it was not feasible to achieve smaller indentations with AFM for very soft samples such as Fmoc-FF gels and cells used here. The trigger force applied in the force volume mode should be large enough to prevent false contact and provide stable force mapping. The problem can’t be solved with the usage of softer cantilevers, since they are more affected by the environmental noises and hydrodynamic drag during the force mapping. Therefore, the lowest possible trigger force (around 0.5-1 nN) resulted in indentations around 500-1500 nm.

Comments 9: Materials & Methods. “Chemical structure (Fig. S1 (A)) and FTIR analysis (…)” (line 385). The full-name needs to be stated the first time that a term appears in the text. Then, the abbreviation should be placed between brackets. This comment should be taken into account for the rest of the main manuscript body text.

Response 9:

This inaccuracy has been corrected for all terms in the text.

Comments 10: “(…) υ is the Poisson ration of the sample (assumed to be 0,5 for the cells and hydrogel)”. (lines 473-474). Why did the authors select a Poisson ratio of 0.5 instead of lower values (e.g. 0.30 or 0.35)? A brief explanation should be furnished in this regard.

Response 10:

A Poisson ratio of 0.5 is generally selected for both hydrogels [Ahearne 2005] and cells [Efremov 2019, Harris 2011], as it was shown that they maintain their volume under load. A brief explanation and references were added into the text.

-Ahearne, M., Yang, Y., El Haj, A. J., Then, K. Y., & Liu, K. K. (2005). Characterizing the viscoelastic properties of thin hydrogel-based constructs for tissue engineering applications. Journal of the Royal Society Interface2(5), 455-463.

-Efremov, Y. M., Velay-Lizancos, M., Weaver, C. J., Athamneh, A. I., Zavattieri, P. D., Suter, D. M., & Raman, A. (2019). Anisotropy vs isotropy in living cell indentation with AFM. Scientific reports9(1), 5757.

-Harris, A. R., & Charras, G. T. (2011). Experimental validation of atomic force microscopy-based cell elasticity measurements. Nanotechnology22(34), 345102.

Comments 11: “5. Conclusions” (lines 525-537). This section perfectly remarks the most relevant outcomes found by the authors in this work and also the promising future prospectives. The authors should furnish a brief statement to discuss about the future action lines to pursue the topic covered in this work.

Response 11:

The section “Conclusions” were was expanded, further plans for work in this area were described, please see the manuscript: “In the future, it is planned to modify the composition of the gel to make it more suitable as a biocompatible scaffold for living cells. This could be achieved, for example, by adding other peptides such as Fmoc-F5 or PEG8-(FY)3 hexapeptide [Diaferia 2019]. Subsequently, specific tissue types can be targeted by using epithelial or neuronal cells (depending on the task at hand). This will allow for the dynamic study of changes in cell metabolism when adhered to the surface of self-assembling gels, the effects of growth factors and other substances on the systems under investigation, and simultaneously, the degradation of the gel itself using the SIСM method”.

-Diaferia, C.; Ghosh, M.; Sibillano, T.; Gallo, E.; Stornaiuolo, M.; Giannini, C.; Morelli, G.; Adler-Abramovich, L.; Accardo, A. Fmoc-FF and hexapeptide-based multicomponent hydrogels as scaffold materials. Soft Matter 2019, 15(3), 487-496.

Round 2

Reviewer 1 Report

Comments and Suggestions for Authors

All questions are resolved and can be published.

Reviewer 2 Report

Comments and Suggestions for Authors

The authors did a great deal of effort to cover all the suggestions raised by the Reviewers. For it, the scientific manuscript quality was greatly improved. Based on the novelty of the gathered results, I warmly endorse this work for further publication in the International Journal of Molecular Sciences.